# Electronically decoupled stacking fault tetrahedra embedded in Au(111) films

Koen Schouteden[1], Behnam Amin-Ahmadi[2], Zhe Li[1], Dmitry Muzychenko[3], Dominique Schryvers[2] & Chris Van Haesendonck[1]

Stacking faults are known as defective structures in crystalline materials that typically lower the structural quality of the material. Here, we show that a particular type of defect, that is, stacking fault tetrahedra (SFTs), exhibits pronounced quantized electronic behaviour, revealing a potential synthetic route to decoupled nanoparticles in metal films. We report on the electronic properties of SFTs that exist in Au(111) films, as evidenced by scanning tunnelling microscopy and confirmed by transmission electron microscopy. We find that the SFTs reveal a remarkable decoupling from their metal surroundings, leading to pronounced energy level quantization effects within the SFTs. The electronic behaviour of the SFTs can be described well by the particle-in-a-box model. Our findings demonstrate that controlled preparation of SFTs may offer an alternative way to achieve well-decoupled nanoparticles of high crystalline quality in metal thin films without the need of thin insulating layers.

[1] Solid-State Physics and Magnetism Section, KU Leuven, BE-3001 Leuven, Belgium. [2] Electron Microscopy for Materials Science (EMAT), University of Antwerp, BE-2020 Antwerp, Belgium. [3] Faculty of Physics, M.V. Lomonosov Moscow State University, 119991 Moscow, Russia. Correspondence and requests for materials should be addressed to K.S. (email: koen.schouteden@fys.kuleuven.be).

Stacking faults (SFs) are defect-type structures that occur in crystalline materials, for example, due to a local mismatch of the atomic stacking within the crystallographic planes or due to a deviation in the stacking sequence of the planes. These defects can occur more frequently in crystalline films that are grown on substrates with different lattice. SFs are typically considered to be undesired defects that lower the film structural properties and therefore much effort is done to avoid their formation[1,2]. However, they also show intriguing electronic properties[3,4] that may be exploited if they can be created in a controlled manner. The formation of SFs in crystals is promoted by quenching from high temperature, by high-energy particle irradiation and by doping with, for example, Mg, Cd or Zn[1,2,5–7]. Alternatively, their formation can also be promoted by growth of thin films on selected substrates with an appropriate lattice mismatch.[3]

One very particular type of defect is the so-called stacking fault tetrahedron (SFT), which consists of four different triangular-shaped SF planes that together demarcate a three-dimensional quasi-perfect nanocrystal. Previously, we reported on lateral quantization effects in SFTs in pristine Ag(111) surfaces grown on mica[4]. These SFTs appear spontaneously during the Ag film growth and they are known to exist in various metals[5–7], including Ag(111) films[3,8]. Thus far, studies have focused on the growth and annihilation of SFs, yet the electronic properties of SFTs have remained largely unexplored.

Here, we report on the electronic properties of SFTs that are retrieved in Au(111) films grown on mica and that we investigate by scanning tunnelling microscopy (STM), scanning tunnelling spectroscopy (STS) and high-resolution transmission electron microscopy (HRTEM). We find that the embedded Au SFTs reveal a remarkable decoupling from their metal surrounding, which can be attributed to their stacking-fault-type origin. This implies that the SFTs may be considered as metallic quantum dots that are embedded in a metallic film. The Au SFTs accommodate an electronic state at their exposed surface that differs strongly from that of the surrounding Au(111) surface state. Au SFTs therefore reveal a very different electronic behaviour than previously investigated Ag SFTs, which showed a lateral quantization effect of the Ag(111) surface state without a clear decoupling from the surrounding Ag(111) substrate[4]. This remarkable difference highlights the rich and diverse electronic properties of SFTs, which appear to strongly depend on the material.

## Results

**Identification of SFTs.** The Au(111) surface is well known for its remarkable surface reconstruction that is commonly referred to as a herringbone reconstruction[9,10]. It consists of a periodic modulation of the surface topography, in which surface atoms rearrange in either face centred cubic (fcc) or hexagonal close packed (hcp) stacking. Hcp and fcc regions are separated by discommensuration lines in which the atoms are slightly squeezed out of the otherwise atomically flat (111) surface. These ridges are running along three directions following the (111) surface and typically switch their orientation in a periodic, herringbone-type manner. At the elbows of the reconstruction ridges, a single atomic point dislocation exists[11,12].

Locally, the herringbone ridges can show a more disordered appearance. In these regions, it is occasionally observed that three pairs of herringbone ridges seem to merge together as illustrated in Fig. 1a. Within our experiments, we find that at 1 to 10% of such crossroads, a larger defect-like feature can exist . These defect-like features have a very regular shape and they either appear as depressions or as protrusions, as is the case for the two highlighted defects in Fig. 1a. At the used tunnelling voltage, the defect enclosed by the dotted circle has a depth of $25 \pm 5$ pm, while the defect enclosed by the dashed circle has a height of $35 \pm 5$ pm. Owing to their exclusive appearance at these herringbone crossroads and their crystalline shapes, these defects are considered to have a similar origin as the SFTs that exist in Ag(111) films[8]. Apart from the regular-shaped defects at the crossroads, also other defects can be observed in Fig. 1a. These defects are commonly observed in STM images of our Au(111) films and are interpreted as (sub)surface atomic-size defects (an impurity atom or a Au vacancy) in the top atomic layers of the Au(111) film. These defects act as effective scattering

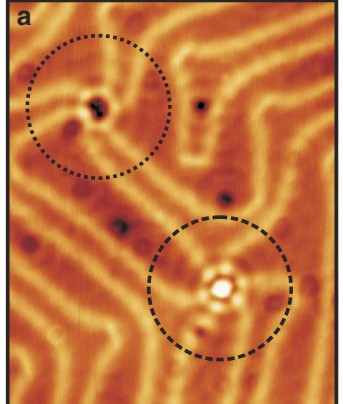
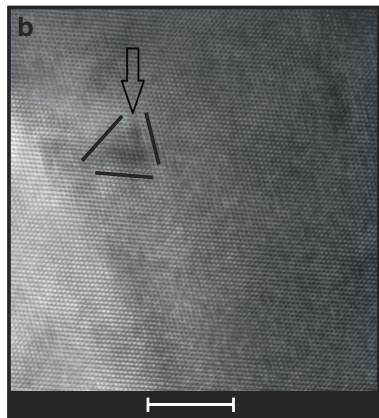
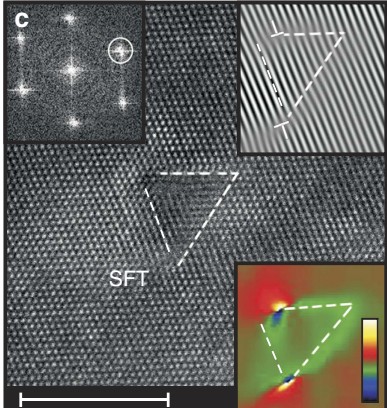

**Figure 1 | Identification of the Au stacking fault tetrahedra. (a)** Scanning tunnelling microscopy topography of two defect-like features with the shape of a truncated triangle ($V = +100$ mV, $I = 1$ nA), which are interpreted as (truncated) SFTs. At the Au(111) surface, these features exist at the crossroads of three pairs of herringbone ridges exclusively. Image size: $29 \times 36$ nm$^2$. **(b,c)** Cross-sectional <110> high-resolution transmission electron microscopy micrographs of the Au(111) film revealing the presence of stacking fault tetrahedra (SFTs), which are highlighted by black solid and white dashed lines, respectively. The top left inset in **c** shows the fast Fourier transform pattern in which the reciprocal lattice vector $\mathbf{g} = \bar{1}11$ is indicated by a white circle. The inverse fast Fourier transform of $\mathbf{g} = \bar{1}11$ is presented in the upper right inset. It shows the shift of the {111} planes owing to the presence of the stacking faults in **c**. The local strain map ($\mathbf{g} = \bar{1}11$) of the indicated SFT is presented in the lower right inset, which shows the dislocations as hot spots and corresponding strain gradients due to the SFT. The scale bars in **b,c** correspond to a length of 5 nm. The colour scale bar in the lower right inset in **c** represents the relative strain variations. The scale bar for strain mapping is between $-12\%$ (black colour) to 12% (white colour).

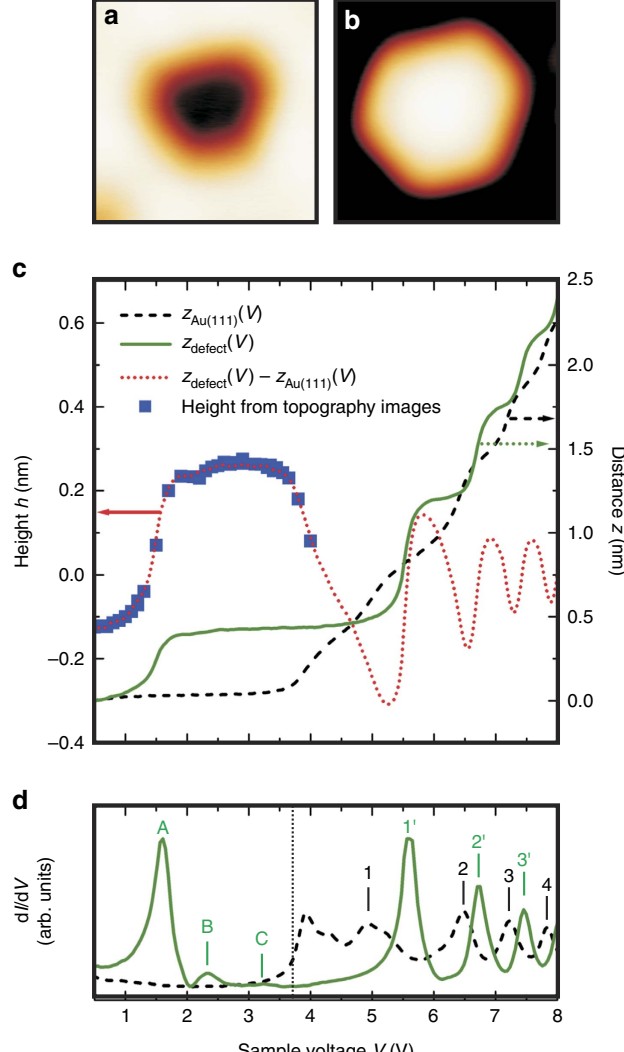

**Figure 2 | Electronic behaviour of the stacking fault tetrahedra.**
(**a**,**b**) Scanning tunnelling microscopy (STM) topography of a SFT different from the ones in Fig. 1a recorded at −1,000 mV and +3,100 mV, respectively. Image sizes: $6 \times 6$ nm². (**c**) Voltage-dependent height/depth (red dotted curve) of the SFT in **a**,**b** determined via $z(V)$ spectroscopy (closed feedback loop) of the SFT (green solid curve) and the surrounding Au(111) (black dashed curve). Blue data points are obtained from STM topographies recorded at different voltages. (**d**) Corresponding d$I$/d$V$ spectra of the Au SFT and the surrounding Au(111) surface (closed feedback loop).

centres for both surface and bulk electrons, which we have previously reported in ref. 13. The existence of SFTs in our gold films is directly confirmed via HRTEM experiments and corresponding inverse fast Fourier transform and local strain mapping (showing local atom displacements due to the presence of the SFT). Two examples of a triangular shaped SFT are presented in Fig. 1b,c, displaying similar contrast as in previous TEM work[2]. The more hexagonal shape of the defects at the Au(111) surface in Fig. 1a points to truncated SFTs[1,14]. In the following, such defects are referred to as SFTs. On the basis of our STM and STS experiments discussed below, we rule out other possible origins for the observed SFTs. We find that the SFTs are retrieved less frequently in Au(111) films when compared with Ag(111) films[4], which can be explained by the fact

that Au(111) films have a larger stacking-fault formation energy[15].

For our experiments, the observed lateral sizes of the Au SFTs are in the 1 to 5 nm range, that is, about two to three times smaller than the Ag(111) SFTs reported in ref. 4. The Au SFTs were observed on nine different samples (each prepared as described in the 'Methods' Section), irrespective of the amount of cleaning cycles (ranging from one up to eight cycles of ion bombardment and annealing). In total, more than 100 SFTs were retrieved, all of them at herringbone crossroads similar to the SFT presented in Fig. 1a. The amount (density) of SFTs can vary considerably from one sample to another (ranging from only one SFT within 1 µm² to several SFTs within 0.01 µm²). We did not find a clear relation between the annealing time (ranging from 1 h to more than 12 h) and the annealing temperature (ranging from 330 to 430 °C) on one hand, and the density of SFTs on the other hand. The observed variation of the SFT densities among our samples may be related to a variation of the density of impurities and vacancies in our gold films, which can act as nucleation centres for SFT formation[7]. The SFT size and density may be tuned by varying the growth parameters (for example, film thickness, involved temperatures, rate of deposition, and so on) and by selecting a different substrate for the Au(111) film growth[3].

**The electronic behaviour of SFTs**. Next, we performed a detailed STM investigation of the SFTs. Remarkably, the height of the SFTs (with respect to the surrounding Au(111) surface) in the STM topographies depends strongly on the tunnelling voltage. This is illustrated in Fig. 2a,b. At voltages below 1 V, the SFT appears as a depression with a voltage-independent depth (Fig. 2a). At voltages above 1V, the depth is strongly voltage-dependent and the SFT can even appear higher than the surrounding Au(111) surface (Fig. 2b). The dependence of the height of the SFT on the tunnelling voltage is demonstrated in more detail in Fig. 2c, which shows the height of the SFT in Fig. 2a,b as inferred from STM topographies (blue dots) and from distance–voltage $z(V)$ curves (red dotted line) that are recorded on the SFT (green solid line) and the surrounding Au(111) surface (black dashed line). Figure 2c demonstrates that the height of the SFT oscillates with increasing voltage. This peculiar behaviour is observed for all thus investigated SFTs and can be accounted for by the specific electronic structure of the SFTs.

Figure 2d presents d$I$/d$V$ spectra that are recorded together with the (black dashed and green solid) $z(V)$ spectra in Fig. 2c. These spectra reflect the local density of states (LDOS) of the SFT and the Au(111) surface. The Au(111) spectrum is more or less featureless within the surface band gap that ranges up to about 3.7 V (ref. 16). Around this voltage, the bottom of the bulk conduction band appears as a pronounced step in the d$I$/d$V$ signal (indicated by vertical black dotted line). At higher voltages, four image-potential states are revealed (labelled 1 to 4). These electronic resonances in Fig. 2d are observed as a step in the corresponding $z(V)$ spectrum in Fig. 2c. Image-potential states exist below the vacuum level, yet they are shifted to higher voltages due to the applied electric field in the STM experiments[17]. Electrons within an image-potential state act as a two-dimensional free-electron-like gas that can move freely parallel to the surface. In contrast to the Au(111) spectrum, the SFT spectrum reveals the presence of pronounced electronic resonances in the same voltage region, that is, around 1.6, 2.3 and 3.2 V (labelled A to C), as well as three image-potential states (labelled 1′ to 3′). Remarkably, the step-like onset of the bulk conduction band is completely absent in the

SFT spectrum. This absence indicates that scattering from SFT electrons to Au bulk states is (nearly) absent and hence points to a strong decoupling of the Au SFT from the surrounding Au(111) surface.

The image-potential states of the investigated Au SFTs always appear at higher voltages compared with the corresponding voltages of the surrounding Au(111) surface, as illustrated in Fig. 2d (also see Supplementary Fig. 1). We note here that it can be excluded that the states labelled A to C are image-potential states, as will be demonstrated below. The voltage difference between the image-potential states of the Au(111) and SFT (1′-1, 2′-2, 3′-3, …) is typically several hundreds of meV (see Supplementary Fig. 1). We can then conclude that the work function of the SFTs is higher than that of the bare Au(111) surface[18]. Following theoretical work reported in ref. 19, this implies that the two-dimensional image-potential states should not be confined within the contours of the Au SFTs, in contrast to Ag SFTs in Ag(111) (ref. 4). The increased work function may be attributed to a decreased distance between the successive atomic layers in the Au SFT compared with the surrounding Au(111) atomic layers[20].

The above-described observations hold for all SFTs that we investigated with STS: (1) absence of Au(111) bulk conduction band, (2) pronounced resonances between 1 and 4 V and (3) image-potential states occur at higher voltages compared with the surrounding Au(111). These common properties of the SFTs, in addition to their similar appearance, imply that the SFTs have a similar structure and hence a similar origin.

**Quantization effects in SFTs.** Next, we focus on the electronic resonances of the SFTs that are resolved at lower voltages, such as

those labelled A to C in Fig. 2d. Figure 3a presents an STM topography of another, larger Au SFT. Corresponding d$I$/d$V$ spectra are presented in Fig. 3b and reveal maxima around 1.6, 2.1 and 2.6 V. The spectra are recorded with the same tunnelling voltage setpoint yet with different tunnelling current setpoints, implying different electric fields between the STM tip and the sample. It can be seen in Fig. 3b that the resonances do not exhibit a detectable Stark shift for the used settings. This excludes interpretation of the resonances in terms of image-potential states, which are strongly dependent on the electric field[17].

Figure 3c displays a selection of LDOS maps recorded on the Au SFT in Fig. 3a. It can be seen that pronounced wave patterns start to develop within the contours of the Au SFT for voltages exceeding about 1.5 V. The patterns within the SFT have a very high intensity compared with that of the surrounding Au(111) surface up to the bottom of the bulk conduction band around 3.7 V (see Fig. 2d). Above this value, the signal on the Au(111) surface increases drastically and wave patterns of the SFT become more and more difficult to discern. Standing waves cannot be observed on the surrounding Au(111) surface above 3.7 V due to the strong coupling of the Au(111) surface state to the bulk states. On the Au SFT, wave patterns can be resolved up to about 4.5 V (more d$I$/d$V$ maps of the Au SFT in Fig. 3a are presented in Supplementary Fig. 2). This further confirms the strong decoupling of the Au SFTs from the surrounding Au(111).

It is clear that the resonances and the wave patterns in Fig. 3c do not exhibit a repeating periodic behaviour with applied tunnelling voltage (also see Supplementary Fig. 2). This excludes interpretation of the Au SFTs in terms of subsurface Ar bubbles

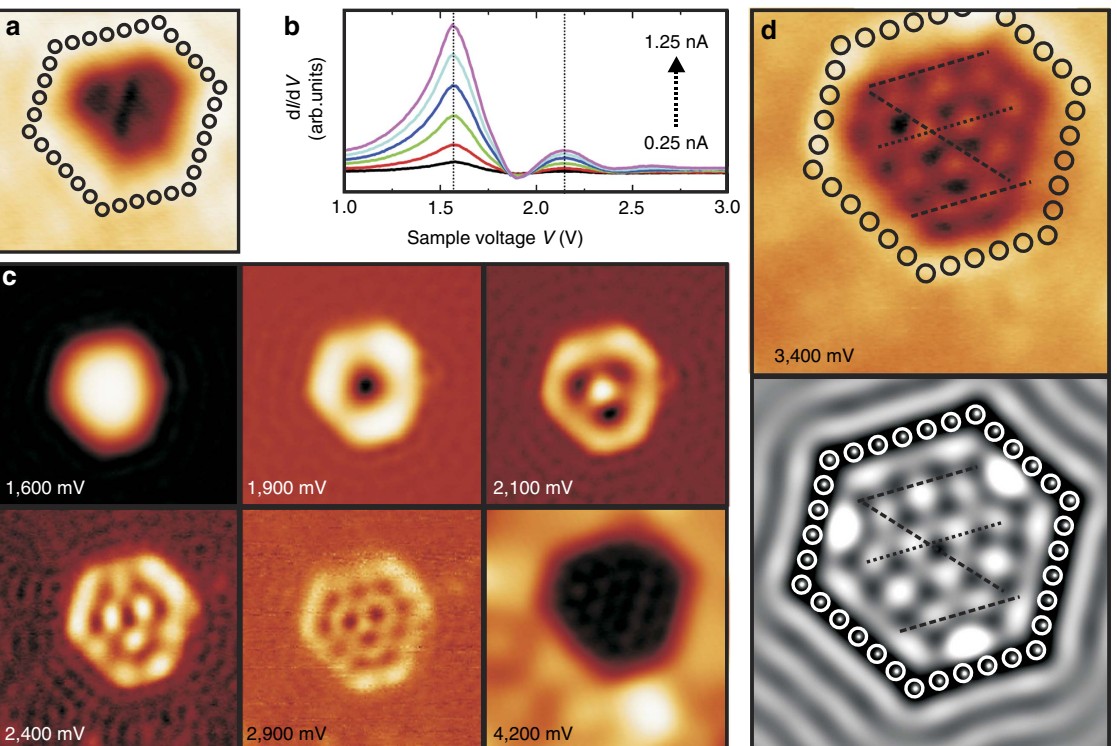

**Figure 3 | Quantization in a larger stacking fault tetrahedron.** (**a**) Scanning tunnelling microscopy (STM) topography of a larger Au SFT ($V = 750$ mV; image size $= 7 \times 7$ nm$^2$). (**b**) Current-dependent d$I$/d$V$ spectra (closed feedback loop) recorded at the centre of the SFT in **a**. The spectra reveal no significant shift of the electronic states with varying electric field, which is determined by the tunnelling current. (**c**) Corresponding series of local density of states (LDOS) maps recorded at the indicated voltages (image sizes are $9 \times 9$ nm$^2$, except for the image at 4,200 mV which is $6 \times 6$ nm$^2$). (**d**) LDOS map at 3,400 mV (top) and simulated image (bottom). Dashed lines are added as guides for the eye. Circles in **a**,**d** indicate the position of the adatom-scatterers that are used to obtain the simulated image in **d** (refs 25,26). Image sizes are $7 \times 7$ nm$^2$.

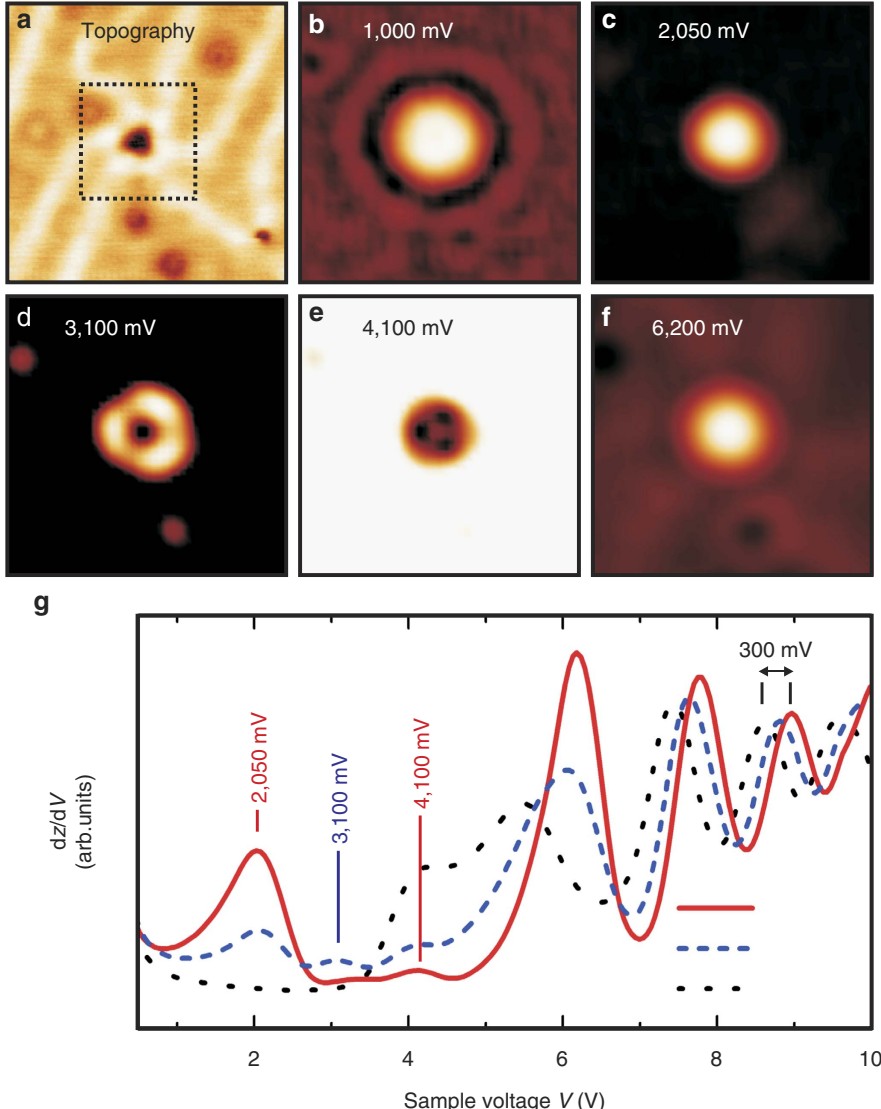

**Figure 4 | Quantization in a smaller stacking fault tetrahedron. (a)** Scanning tunnelling microscopy (STM) topography of a small Au SFT. Image size: $16 \times 16 \, nm^2$. $V = -250 \, mV$, $I = 1 \, nA$. **(b–f)** Corresponding $dz/dV$ maps of the region indicated by the dotted square in **a** at the indicated voltages. Image size: $6.5 \times 6.5 \, nm^2$. **(g)** $dz/dV$ spectra recorded at the centre of the maximum in **c** (red solid curve) and at the centre of the area enclosed by the three maxima in **d** (blue dashed curve). A spectrum of the surrounding Au(111) is added as a reference (black dotted curve).

that may remain after cleaning of the sample. Scattering of bulk electrons between the subsurface Ar bubble and the metal film surface leads to quantum-well-type resonances and electron standing waves that show a periodic behaviour with applied tunnelling voltage[21,22]. Given the applied high annealing temperature, the presence of remaining Ar bubbles in our Au(111) films is unlikely. Moreover, in the case of subsurface Ar bubbles, one expects to still probe the Au(111) bulk conduction band in $dI/dV$ spectra recorded above the bubble. The Au SFTs have an appearance that is similar to that of Au vacancy islands that can be controllably created by mild ion bombardment and annealing[23]. However, while vacancy islands have a depth of one atomic layer that is independent of the applied tunnelling voltage, Au SFTs have a sub-monolayer depth/height that depends strongly on the applied tunnelling voltage as discussed above. Moreover, the electronic behaviour of Au SFTs differs strongly from that of Au vacancy islands[23,24]. Au vacancy islands confine the surface state of the Au(111) surface within their step boundaries. The surface state electrons

within the vacancy islands experience a considerable coupling to the bare Au(111) substrate and their confinement persists only up to the onset of the bulk conduction band around 3.7 eV.

We therefore interpret the observed wave patterns and energy resonances of the Au SFTs as a new electronic state that exists within the Au SFTs and that can be probed at the SFT facet that is exposed at the surface. This electronic state of the SFT exhibits a behaviour reflecting that of a surface state. The known surface state of the Au(111) surface is characterized by a parabolic-like dispersion with an onset energy $E_0 = -460 \, meV$ and an effective electron mass $m^\star = 0.23 \, m_e$ (refs 13,24). To learn more about the electronic state of the Au SFTs, we performed simulations using the particle-in-a-box software (available via ref. 25) developed by K.-F. Braun[26]. For the Au SFT in Fig. 3, we achieve good agreement between the experimental LDOS maps and the 2D particle-in-a-box model when using an onset energy $E_0 = 1,490 \, meV$ and an effective electron mass $m^\star = 0.33 \, m_e$. Figure 3d presents an experimental LDOS map together with the result of the simulation at 3,400 mV. More simulation images

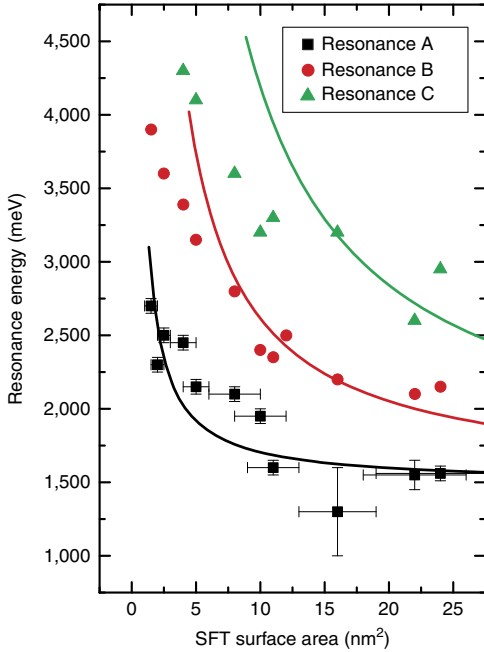

**Figure 5 | Size dependence of the quantization effects.** Energy (voltage) values of the electronic resonances that are observed for Au stacking fault tetrahedra (SFTs) of different size. The area of the exposed SFT facet at the Au(111) surface is determined from scanning tunnelling microscopy topographies. The solid lines are plots of the theoretically expected size dependence for a hexagonal box following equation (1) using $E_0 = 1,490$ meV and $m^* = 0.33\ m_e$ (ref. 23). For each SFT surface area, the error bars for resonances B and C are the same as the ones for resonance A. The error bar for the determined surface area scales with the square of the diameter of the exposed SFT surface area, where the error on the diameter is determined by the tip convolution effects. The error for the resonance energy is determined by the used voltage modulation (if determined from scanning tunnelling spectroscopy measurements) or by the used voltage interval (if determined from voltage-dependent local density of states mappings as is the case for the data point at 16 nm²).

are presented in Supplementary Fig. 2. These used values for $E_0$ and $m^*$ differ strongly from those of the Au(111) surface state, which may be accounted for by a different stacking of the Au atoms within the SFT when compared with the Au(111) film, as indicated above.

**Comparison with the particle-in-a-box model.** To further verify our interpretation in terms of quantum confinement within the Au SFTs, we now focus on a Au SFT that is significantly smaller. The lateral size (exposed facet) of the SFT in Fig. 4a is only about 1.5–2.0 nm. Corresponding dz/dV maps that reflect the LDOS are presented in Fig. 4b–f. The dz/dV spectrum in Fig. 4g reveals the existence of three electronic resonances below the voltage at which the first image-potential state occurs, that is, at 2,050, 3,100 and 4,100 mV. Here as well, the image-potential states above the SFT occur at higher voltages compared with the surrounding Au(111) surface. The dz/dV maps at 2,050, 3,100 and 4,100 mV are similar to the maps for the larger SFT in Fig. 3 at 1,600, 1,900 and 2,100 mV, respectively.

The three electronic resonances of the small SFT exist at higher energies compared with the larger SFT in Fig. 3, in agreement with our interpretation in terms of the particle-in-a-box model. Considering the same onset energy $E_{0,SFT}$ and effective electron mass $m^*$, the electronic resonances $E_{n,SFT}$ (eigenstates)

and the corresponding wave patterns of SFTs with different size $\Omega$ can be linked to each other using the particle-in-a-box equation

$$E_{n,SFT} = E_{0,SFT} + \lambda_{n,SFT} \times (m^* \times \Omega)^{-1}, n = 1, 2, 3, \ldots \quad (1)$$

In equation (1), the ;eigenvalues $\lambda_{n,SFT}$ depend solely on the shape of the confining box. Assuming the same (truncated triangular) shape for the SFT in Fig. 4 as that used to model the data in Fig. 3, we find good agreement between the simulated images and the experimental dz/dV maps if $\Omega$ is taken 25% the size of that used for Fig. 3. This is in very good agreement with the size determined based on the STM topography in Fig. 4a.

The dI/dV maps of the Au SFT in Fig. 2a,b are presented in Supplementary Fig. 3. Maps of yet another Au SFT are presented in Supplementary Fig. 4. This SFT has a very similar lateral size and shape as that in Supplementary Fig. 3 and shows quasi-identical wave patterns and voltage-dependent behaviour. The SFT in Fig. 3 and Supplementary Fig. 2 is slightly larger than the two SFTs in Supplementary Fig. 3 and Supplementary Fig. 4 and similar wave patterns are formed at somewhat lower voltages (for example, the wave pattern at 3,200 mV in Supplementary Fig. 3 and Supplementary Fig. 4 occurs at 2,900–3,000 mV for the NC in Supplementary Fig. 2), again in agreement with the particle-in-a-box model.

Figure 5 presents an overview of the energy values of the electronic resonances of all investigated Au SFTs as a function of their surface area. It can be seen that there exists a strong correlation between the energies and the SFT surface area. This is again in line with interpretation of the electronic resonances in terms of the particle-in-a-box model, that is, the electronic behaviour can be described by the same onset energy $E_0 = 1,490$ meV and the same effective electron mass $m^* = 0.33\ m_e$ for all SFTs. In turn, this additionally indicates that all investigated SFTs have the same atomic structure, that is, they are all (truncated) Au SFTs that occur spontaneously in the Au(111) film and of which one facet is exposed at the Au(111) surface.

Deviations of the experimental data in Fig. 5 from the theoretical model may be attributed at least partially to deviations from the assumed idealized hexagonal (truncated triangular) shape. In particular, deviations in Fig. 5 are most pronounced for the smaller SFTs such as the one in Fig. 4a. For these smaller SFTs, the precise shape can be observed less clearly in STM images and it may be more close to that of a triangle rather than a hexagon. In addition, electron scattering at the subsurface SF planes of the SFT may affect the ideal particle-in-a-box type confinement of electrons.

Finally, as indicated above, the exposed surfaces of the Au SFTs all have a regular crystalline shape, that is, the shape of a hexagon or truncated triangle. Exceptionally, an equilateral, triangularly shaped defect-like feature is found (only one observation, see Supplementary Fig. 5). The arrangement of the herringbone ridges at the triangular shaped feature differs from that of the other SFTs (Supplementary Fig. 5a). Moreover, the triangular feature exhibits an electronic behaviour that differs from the other SFTs. It shows the Au(111) bulk conduction band similar to the surrounding Au(111) surface (Supplementary Fig. 5k). In addition, wave patterns can already be observed in LDOS maps at voltages close to the Fermi level (Supplementary Fig. 5f,g), while the SFTs discussed above reveal wave patterns only above 1,490 mV. The resolved wave patterns (Supplementary Fig. 5e–i) and resonance in the STS spectrum (Supplementary Fig. 5j) can be interpreted as confinement of the bare Au(111) surface state (Supplementary Fig. 5j) following the particle-in-a-box model with $E_0 = -460$ meV and $m^* = 0.23\ m_e$, similar to the results reported in ref. 23 for Au islands. This triangular feature therefore must have a different structure

than the other Au SFTs. The triangular feature may be interpreted as a so-called Frank loop, consisting of a single stacking fault. The Frank loop is very similar to the SFT in terms of the spatial coordinates of atoms.

In conclusion, we performed a detailed STM investigation of Au SFTs that exist in Au(111) films, as confirmed by TEM experiments. The SFTs exhibit a set of discrete electronic resonances and reveal pronounced voltage-dependent wave patterns in maps of the density of states. The wave patterns exist up to energies well above the bottom of the bulk conduction band of the Au(111) film, indicative of a strong decoupling of the Au SFT from its surroundings. This behaviour is found to correlate with the size of the Au SFTs. From analysis using a two-dimensional particle-in-a-box model, we find that the electronic behaviour can be described well by an electronic state with parabolic dispersion having an onset energy of about 1,490 meV above the Fermi level and an effective electron mass of about 0.33 $m_e$.

Our findings demonstrate that controlled introduction of SFTs may offer an alternative way to obtain well-decoupled quantum dots of high crystalline quality in metal thin films without the need of thin insulating layers, which often make sample preparation more cumbersome[27]. In addition, SFTs can be expected to have an enhanced stability at room temperature when compared with deposited nanoclusters of similar size[27,28]. Obviously, their controlled preparation will be a crucial issue for further developments in this direction. A potential route to overcome this issue could be by creating regular patterns of defects in the substrate, at which SFTs may preferentially start to form during the metal film growth on the support. However, it is already evident from our present findings that investigating the electronic properties of SFTs provides a new playground for in-depth studies of quantum mechanical finite size effects in surfaces.

## Methods

**Sample preparation.** Epitaxially grown, 140 nm thick Au(111) films on freshly cleaved mica were prepared by molecular beam epitaxy at elevated temperatures as described in ref. 29. Sample transfer from the molecular beam epitaxy set-up to the low-temperature ultra-high vacuum STM set-up was performed under ambient conditions. The Au(111) surfaces were cleaned in the preparation chamber of the STM set-up by repeated cycles of Ar ion bombardment (at about 4 keV and $10^{-6}$ mbar Ar partial pressure) and annealing (at about 720 K). The resulting film surfaces consist of atomically flat islands with dimensions up to $500 \times 500$ nm$^2$ (ref. 13).

**STM experiments.** All the experiments were conducted in a ultra-high vacuum system (for sample preparation, base pressure in the $10^{-9}$ mbar range) that is connected to a low-temperature STM (Omicron Nanotechnology) operated at 4.5 K (for sample measurement, base pressure in the $10^{-11}$ mbar range). $(dI/dV)(V)$ spectra and $dI/dV$ maps (commonly referred to as LDOS maps) were acquired by lock-in detection with closed feedback loop (amplitude is typically about 20 to 50 mV) at 800 Hz. $(dz/dV)(V)$ spectra and $dz/dV$ maps that also reflect the LDOS are obtained numerically from recorded $z(V)$ spectra. STM data in this work were obtained with mechanically cut PtIr (10% Ir) STM tips, and with polycrystalline W tips that were electrochemically etched and cleaned in situ by thermal treatment. All bias voltages mentioned are with respect to the sample, and the STM tip was virtually grounded. The STM images were analysed using the Nanotec WSxM software[30].

**TEM experiments.** The cross-sectional TEM thin foils were fabricated in a dual beam Helios NanoLab 660 (FEI) setup using the lift-out procedure. To protect the surface of the Au film from damage caused by the incident Ga$^+$ ions of the focused ion beam (FIB)[31], a protective Pt layer was first deposited using electron-beam assisted deposition (5 kV, 0.8 nA) followed by an ion-beam assisted deposited Pt layer (30 kV, 0.23 nA). To minimize any damage on the sample during thinning, final cleaning on both sides of the thin lamella was performed using a low energy ion beam of 1 kV and 95 pA. The HRTEM characterizations of the Au films were carried out using a FEI Tecnai G2 (FEG, 200 kV). To achieve clear visualization of single dislocations and SFs, local strain mapping was performed using the Geometric Phase Analysis, which is an image processing technique that is sensitive to small displacements of the lattice fringes in HRTEM images[32]. Energy-dispersive X-ray analysis was performed in TEM and did not reveal any trace of Ga$^+$ ions in the TEM sample. However, with a detection limit of around 1 at.%, this does not exclude the existence of some Ga in the film and thus the production of extra vacancies, so in order to confirm that SFTs are intrinsic to the present Au films and not FIB induced artifacts, a bulk pure Au (99.99%) reference sample was annealed at 973 K for 24 h. Next, a cross-sectional FIB sample was produced using the same conditions as for the Au film. HRTEM investigation on this sample only revealed dislocation loops and individual SFs, while no SFTs were found (also see Supplementary Fig. 6).

**Calculations.** Simulations were performed using particle-in-a-box software (available via ref. 25) developed by K.-F. Braun[26] (the Schrödinger equation was solved by treating scattering centres at the box boundaries as zero-range potentials).

**Data availability.** All relevant data related to this manuscript are available from the authors.

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

## Acknowledgements

The research in Leuven has been supported by the Research Foundation–Flanders (FWO, Belgium), and by the Flemish Concerted Research Action program (BOF KU Leuven, Project No. GOA/14/007). Z.L. acknowledges additional support from the China Scholarship Council (No. 2011624021). K.S. acknowledges additional support from the FWO. The research in Moscow has been supported by grants of the Russian Foundation for Basic Research (RFBR).

## Author contributions

K.S., D.M. and Z.L. prepared the samples and performed the tunnelling microscopy measurements. B.A.-A. and D.S. performed the electron microscopy measurements. K.S. performed the simulation and wrote the first version of the manuscript. All the authors discussed the results and participated in writing the manuscript.

## Additional information

**Competing financial interests:** The authors declare no competing financial interests.

