## [Peer Review File · Nature Communications]

Reviewers' comments:

Reviewer #1 (Remarks to the Author):

The manuscript by Schouteden et al. reports a unique electronic property of nanometer-sized stacking-fault tetrahedron (SFT), which is a pyramidal-shaped crystal defect, composed of four stacking faults crystallographically equivalent each other. The majority of their SFTs are imperfect pyramid whose apexes are truncated. They demonstrated that the truncated SFTs formed in epitaxially-grown films exhibit a set of discrete electronic resonances. Since they have already reported elsewhere on such a lateral quantization effect of defect clusters formed at the free surface, the highlight of this manuscript is the discovery of this property for defect clusters formed at the interior of films.

The proposed new concept that defect clusters which are generally undesired in materials engineering can be utilized as quantum dots is striking for the research community of crystal defects. I am basically positive for acceptance; however, there are some unclear points whose validity is critical for the highlight of this manuscript. I recommend revising the manuscript in terms of the followings prior to the decision.

[Major comments]

(1) Terminology

In a general terminology, stacking faults are not considered as grain boundaries. Grain boundaries are borders terminating the long-range periodicity of atoms, whereas stacking faults are disorders confined within one atomic layer. Their terminology is not acceptable for the specialists of crystal defects.

(2) FIB and HR-TEM

The detailed conditions of FIB milling are not provided. It is well known that FIB's Ga irradiation introduces vacancies and their clusters, especially in very thin portions sufficient for HR-TEM observation. Those who are aware of the potential artifact of FIB damage never forget to describe the detailed experimental conditions. What they have observed by HR-TEM is most likely the FIB artifact. If that is the case, this manuscript loses its major highlight. A special treatment must be conducted for removing the surface FIB damage. In general, the so-called 'flash electro-polishing' is adopted for the purpose of complete removal of surface damage [1]. The flash polishing is, however, technically rather difficult for non-specialists of electro-polishing. I recommend the authors to approach this issue by using a different method. Observe the microstructure of FIBed samples picked up from a fully annealed Au. If SFTs are not observed, they can validate their results, though this approach is not ideal.

[1] K. Yabuuchi, R. Kasada, A. Kimura, Effect of Mn addition on one-dimensional migration of dislocation loops in body-centered cubic Fe. *Acta Materialia* 61, (2013) 6517-6523.

(3) Simulation

They also mentioned that a triangular 'particle' exhibited a different electronic behavior from the truncated SFTs. Due to the absence of quantum behavior, the authors speculated that the triangular defect is not perfect SFT but a 2D defect, which is the so-called Frank loop, consisting of a single stacking fault. The Frank loop is very similar to the SFT in terms of the spatial coordinates of atoms. In a 2D view from the $\langle 111 \rangle$ directions, the atomic coordinates of the triangular Frank loop are, in theory, exactly the same as those of the perfect SFT. The result of simulation of standing waves based on the 2D particle-in-a-box model for the Frank loop is expected to become the same as that if the

SFT.

(4) Others

The reference style is different from the standard Nature style. The guideline should be respected. English needs to be polished throughout the manuscript. Professional language editing is highly recommended.

[Minor comments]

(1) P2.L44. "in either fcc or hcp stacking"

→ "in either face centered cubic (fcc) or hexagonal close packed (hcp) stacking"

(2) P3.L51-52. "We find ... to exist."

→ The meaning of this sentence is unclear.

(3) P3.L59 and Fig. 1 caption. "local g-mapping"

→ This is less common terminology. "Local strain map" is probably a better expression.

(4) P5.L100. "LDOS"

→ In P14.L270 the authors described " $dI/dV(V)$ spectra and dI/dV maps (referred to as LDOS maps hereafter)"; however, the abbreviation of LDOS is still unclear.

(5) P7.L59. "composition"

→ The observed common properties of the 'NCs' are nothing relevant to composition.

(6) P9.L170. "We also want to ~"

→ This is inappropriate expression in professional scientific papers.

(7) P9.L171. "the here"

→ The meaning is unclear.

Reviewer #2 (Remarks to the Author):

This is a beautiful work based on the state-of-the-art measurements using scanning tunneling microscopy and spectroscopy (STM and STS) and a transmission electron microscope (TEM).

Manuscript is very well organized and clearly written, also references are well selected. Results are clearly showing that stacking faults (STFs) work as effective potential barriers and that the region surrounded by STFs exhibit properties characteristic as an isolated nanocluster, even though the region is embedded in the film of the same material, such as a quantum confinement of electrons. These are very interesting features and, in this manuscript, newly investigated aspects are in particular about the electronic structures of such nanoclusters isolated by STFs.

However, the results are readily understood from previous publications by other groups and even by some of the authors of this manuscript. Therefore, the contents involved in this manuscript are not sufficiently impactful to advance knowledge in this field, thus the manuscript does not meet a criterion of Nature Communications.

I suggest two options. Considering high quality of their experiments, submitting the manuscript to other journals that are more focused on specific topics such as nanomaterials and nanotechnology

would be recommended. Another option is to resubmit the manuscript after involving experimental proof of an impactful consequence of the embedded nanocluster formation, for example, drastic improvement in stability and efficiency in catalytic activities of Au films owing to the formation of embedded nanoclusters.

Reviewer #3 (Remarks to the Author):

The authors studied the electronic properties of stacking fault nanocrystals on Au(111) thin films. Au(111) thin films consisting of atomically flat islands were prepared by molecular beam epitaxy followed by repeated Ar ion bombardment and annealing. From detailed TEM, STM and STS characterizations, the nanocrystals were found to show a set of discrete electronic resonances and display pronounced voltage-dependent wave patterns in the maps of the density of states. The wave patterns exist up to the energies above the bottom of the bulk conduction band of the Au(111) film, indicating that the electronic properties of the Au nanocrystals are decoupled from its surrounding. This behavior was also found to correlate with the size of the Au nanocrystals. The experimental results were further analyzed by using a two-dimensional particle-in-a-box model, which matches well with their experiments.

In general, this paper is well organized and convincing. The topic and the results are novel in this field and will be helpful for understanding the electronic properties of metal nanostructures. Therefore, I would recommend its publication on Nature Communications after some minor revisions.

Below are some detailed comments.

1. The investigation methods and results of this work look quite similar to those for Ag SFTs on Ag(111) (a previous work from the authors, 2012 Phys. Rev. Lett. 108, 076806). I suggest the authors to emphasize the novelty and significance of this work, comparing with their previous work, in the introduction part.
2. What is the difference or relationship between the two-dimensional image potential states reported here and the multipolar plasmonic states reported in larger crystalline nanoprisms (2013 Nat. Mater. 12, 426)?
3. In Figure 5, why are the error bars of the resonances B and C not shown?
4. In Figure 1a, besides the two indicated defects that appear at "crossroads", there are other defects that are not located at the "crossroads". For the HRTEM image, there are no evidences to show that the defect observed on the TEM image is the defect at "crossroads".
5. In Figure 5, the theoretical plots are obviously deviated from the experimental results, especially for the resonances B and C. The authors had better give some explanation for it.
6. The dI/dV spectra of the Au NCs with the triangular shape are clearly seen to be different from those of the Au NCs with the hexagonal shape. The authors should provide discussion on this difference.
7. I do not know the potential applications of these electrically decoupled stacking fault nanocrystals embedded in Au(111) films. Could the authors provide some discussion on this aspect.

The manuscript by Schouteden et al. reports a unique electronic property of nanometer-sized stacking-fault tetrahedron (SFT), which is a pyramidal-shaped crystal defect, composed of four stacking faults crystallographically equivalent each other. The majority of their SFTs are imperfect pyramid whose apexes are truncated. They demonstrated that the truncated SFTs formed in epitaxially-grown films exhibit a set of discrete electronic resonances. Since they have already reported elsewhere on such a lateral quantization effect of defect clusters formed at the free surface, the highlight of this manuscript is the discovery of this property for defect clusters formed at the interior of films.

The proposed new concept that defect clusters which are generally undesired in materials engineering can be utilized as quantum dots is striking for the research community of crystal defects. I am basically positive for acceptance; however, there are some unclear points whose validity is critical for the highlight of this manuscript. I recommend revising the manuscript in terms of the followings prior to the decision.

We are grateful to the reviewer for his/her appreciation of our work and for his/her remarks that help us to further improve the manuscript. Below we reply in detail to his/her remaining concerns. We hope that the reviewer will find our revised manuscript suitable for publication in *Nature Communications*.

[Major comments]

(1) Terminology

In a general terminology, stacking faults are not considered as grain boundaries. Grain boundaries are borders terminating the long-range periodicity of atoms, whereas stacking faults are disorders confined within one atomic layer. Their terminology is not acceptable for the specialists of crystal defects.

We agree that our terminology was not appropriate and have taken into account the comment of the reviewer. In the revised manuscript, we do no longer refer to the SFTs as “particles” or “nanocrystals”. Also we have carefully checked our manuscript upon use of the term “boundaries” and have rephrased wherever required.

(2) FIB and HR-TEM

The detailed conditions of FIB milling are not provided. It is well known that FIB's Ga irradiation introduces vacancies and their clusters, especially in very thin portions sufficient for HR-TEM observation. Those who are aware of the potential artifact of FIB damage never forget to describe the detailed experimental conditions. What they have observed by HR-TEM is most likely the FIB artifact. If that is the case, this manuscript loses its major highlight. A special treatment must be conducted for removing the surface FIB damage. In general, the so-called 'flash electro-polishing' is adopted for the purpose of complete removal of surface damage [1]. The flash polishing is, however, technically rather difficult for non-specialists of electro-polishing. I recommend the authors to approach this issue by using a different method. Observe the microstructure of FIBed samples picked up from a fully annealed Au. If SFTs are not observed, they can validate their results, though this approach is not ideal.

[1] K. Yabuuchi, R. Kasada, A. Kimura, *Effect of Mn addition on one-dimensional migration of dislocation loops in body-centered cubic Fe. Acta Materialia* 61, (2013) 6517-6523.

Firstly, we agree with the reviewer that the details of the FIB TEM sample preparation should be given in the paper. We believe we have a good track record in using FIB for TEM sample preparation in alloys, which has led to several publications focusing on the improvement of the thinning technique and the interpretation of the induced defects [1-3]. Our interpretation that the observed SFT are not due to the FIB thinning in part stems from this experience. Of course, the fact that the SFTs were also observed using the central technique of the paper, the STM technique (Fig. 1a), it is not unexpected that the SFTs are also observed using HRTEM (FIB sample), as they are a confirmed and intrinsic feature of the Au films. This is also part of the reason that the technical details of the TEM work were less explicitly stated in the first version of the manuscript.

In view of our experience, we did take special caution during FIB preparation in order to avoid any surface FIB damage. Electron-beam assisted Pt deposition on the surface of the Au film was first applied with very low energy electrons (5 kV, 0.8 nA) in order not to damage the surface, followed by an ion-beam assisted Pt layer deposition (30 kV, 0.23 nA) which totally prevents the penetration of Ga⁺ ions in the top surface of the Au film. After the thinning steps, a final cleaning on both sides of the thin lamella was performed using a very low energy ion beam of 1 kV and 95 pA. We also performed energy dispersive X-ray (EDX) analysis in TEM and no trace of Ga was detected. Therefore, we do not expect any top- nor side-surface FIB damage in our Au films. Following the remark of the reviewer, the details of the thinning and cleaning procedures are now added to the Methods section of the manuscript and we hereby added the reference indicated by the reviewer.

Secondly, flash polishing is indeed a potential way for further cleaning the damaged FIB samples and which polishes a few to a few dozen nanometers of the specimen surface. This technique has mainly been applied for polishing steel samples [4-6], but Au, being a noble metal, will probably be hard to electropolish (*e.g.*, the Fischione polishing recipes do not show an entry for pure Au). In other work we, so far unsuccessfully, attempted some flash-polishing of Ni FIB samples and, as the reviewer mentions, application of this technique is technically rather difficult.

Thirdly, annealing of our Au films could be a good alternative, yet in our work it is unfortunately not possible, due to the presence of the underlying mica which risks to react with the Au at high temperatures. As described in Ref. [Vand2003], during the growth of the Au(111) film on mica in UHV, the sample is exposed to temperatures of 530 °C. This sample growth procedure yields Au(111) films with large atomically flat terraces and a low density of SFTs, as reported in the manuscript. In Au(111) single crystals or in films grown on other substrates, the density of SFTs can obviously be different. It is known that, depending on the lattice mismatch, growth of thin metal films on a substrate may yield a higher density of stacking faults compared to that in a single crystal [Uchi20120]. The density of SFTs in our films on mica may be varied by tuning the growth parameters (film thickness, involved temperatures, rate of deposition *etc.*). Previously we attempted UHV annealing experiments of Au(111) films on mica to higher temperatures of about 750 °C. This annealing gave rise to severe degradation of the Au(111) film. For this reason, we cannot apply meaningful annealing to higher temperatures for the present work. In the revised manuscript, we comment that the SFT size and density may be tuned by varying the growth parameters (*e.g.*, film thickness, involved temperatures, rate of deposition, *etc.*) and by selecting a different substrate for the Au(111) film growth (see, *e.g.*, Uchihashi, T., Kobayashi, K. & Nakayama, T. *Phys. Rev. B* **82**, 113413 (2010)).

Finally, in line with the suggestion of the reviewer, another FIB sample from a different Au deposited film [polycrystalline gold layer (around 250 nm) deposited on a perovskite film] was prepared with exactly the

same FIB preparation procedure and conditions explained in the Methods section of the manuscript. A typical HRTEM image of this sample is shown in the below figure and does not show any SFT at areas with some blurring, probably due to the FIB-based sample preparation. Therefore, it can be concluded that the observed SFTs (Fig. 1b) in the manuscript are indeed an intrinsic feature of the current Au(111) films on mica, as was also the case for previously investigated Ag(111) films grown on mica [Scho2012].

[1] Reducing the formation of FIB-induced FCC layers on Cu-Zn-Al austenite, Zelaya E., Schryvers D., *Microsc Res Techniq* 74(1), 84 (2011).

[2] Evaluation of top, angle, and side cleaned FIB samples for TEM analysis, Montoya E., Bals S., Rossell M. D., Schryvers D., van Tendeloo G., *Microsc Res Techniq* 70(12), 1060 (2007).

[3] Point defect clusters and dislocations in FIB irradiated nanocrystalline aluminum films : an electron tomography and aberration-corrected high-resolution ADF-STEM study, Idrissi H., Turner S., Mitsuhashi M., Wang B., Hata S., Coulombier M., Raskin J.-P., Pardoën T., van Tendeloo G., Schryvers D., *Microsc Microanal* 17(6), 983 (2011).

[4] M. Ando et al., *Journal of Nuclear Materials* 271 & 272 (1999) 111–114.

[5] K. Fujii, K. Fukuya, *Journal of Nuclear Materials* 336 (2005) 323–330

[6] K. Yabuuchi et al., *Acta Materialia* 61 (2013) 6517–6523

(3) Simulation

They also mentioned that a triangular 'particle' exhibited a different electronic behavior from the truncated SFTs. Due to the absence of quantum behavior, the authors speculated that the triangular defect is not perfect SFT but a 2D defect, which is the so-called Frank loop, consisting of a single stacking fault. The Frank loop is very similar to the SFT in terms of the spatial coordinates of atoms. In a 2D view from the <111> directions, the atomic coordinates of the triangular Frank loop are, in theory, exactly the same as those of the perfect SFT. The result of simulation of standing waves based on the 2D particle-in-a-box model for the Frank loop is expected to become the same as that of the SFT.

We thank the reviewer for pointing us to a possible interpretation of our observed triangular defect-like feature. We have added this interpretation to the related paragraph in our manuscript.

Furthermore, we believe that our discussion of its electronic properties was somewhat confusing. In particular, the triangular defect also exhibits quantum behaviour. The standing waves that are observed within the defect as well as the STS spectra recorded at the defect can be explained by quantum confinement of the Au(111) surface state within the defect. As the reviewer indicates, this indeed is in line with the 2D particle-in-a-box model, *i.e.*, when considering an onset energy and effective mass of the confined electrons that is the same as that for the Au(111) surface state ($E_0 = -460$ meV and $m^* = 0.23 m_e$). In this respect, the triangular defect shows a quantum confinement behaviour that is quasi-identical to that of Au islands and Au vacancy islands that can be created by mild ion bombardment.

In our manuscript we want to emphasize that the quantum confinement behaviour of the triangular defect is very different from that of the truncated triangular defects (SFTs) that exist at the crossroads of three herringbones exclusively. In particular, modelling of the SFT quantum behaviour using the 2D particle-in-a-box model reveals a very different onset energy and electron effective mass, *i.e.*, $E_0 = 1490$ meV and $m^* = 0.33 m_e$. Moreover, the triangular defect does reveal the onset of the Au(111) bulk conduction band, in contrast to the (truncated) SFTs. In the revised manuscript, we have rephrased the related paragraphs to better emphasize our reasoning.

(4) Others

The reference style is different from the standard Nature style. The guideline should be respected. English needs to be polished throughout the manuscript. Professional language editing is highly recommended.

In the revised version of the manuscript we have modified the style of our references following the guidelines of *Nature Communications*.

[Minor comments]

(1) P2.L44. "in either fcc or hcp stacking"

→ "in either face centered cubic (fcc) or hexagonal close packed (hcp) stacking"

In the revised manuscript we define the abbreviations fcc and hcp upon their first use.

(2) P3.L51-52. *"We find ... to exist."*

→ *The meaning of this sentence is unclear.*

We have rephrased the indicated sentence to enhance its clarity.

(3) P3.L59 and Fig. 1 caption. *"local g-mapping"*

→ *This is less common terminology. "Local strain map" is probably a better expression.*

We followed the suggestion of the reviewer and now use the term “local strain map” instead of “local g-mapping”.

(4) P5.L100. *"LDOS"*

→ *In P14.L270 the authors described "(dI/dV)(V) spectra and dI/dV maps (referred to as LDOS maps hereafter)"; however, the abbreviation of LDOS is still unclear.*

In the revised manuscript we define the abbreviation of “LDOS” at first use.

(5) P7.L59. *"composition"*

→ *The observed common properties of the 'NCs' are nothing relevant to composition.*

By “composition” we were referring to the fact that the SFTs should be composed of the same chemical element, *i.e.*, gold. However, since this may be considered as trivial and to avoid possible confusion, we have removed the word “composition” throughout the text.

(6) P9.L170. *"We also want to ~"*

→ *This is inappropriate expression in professional scientific papers.*

We have removed the expression from the manuscript and rephrased the sentence.

(7) P9.L171. *"the here"*

→ *The meaning is unclear.*

We have rephrased the sentence to improve its clarity.

Our reply (in green) to the remarks (*italic*) of Reviewer #2

This is a beautiful work based on the state-of-the-art measurements using scanning tunneling microscopy and spectroscopy (STM and STS) and a transmission electron microscope (TEM).

Manuscript is very well organized and clearly written, also references are well selected. Results are clearly showing that stacking faults (STFs) work as effective potential barriers and that the region surrounded by STFs exhibit properties characteristic as an isolated nanocluster, even though the region is embedded in the film of the same material, such as a quantum confinement of electrons. These are very interesting features and, in this manuscript, newly investigated aspects are in particular about the electronic structures of such nanoclusters isolated by STFs.

However, the results are readily understood from previous publications by other groups and even by some of the authors of this manuscript. Therefore, the contents involved in this manuscript are not sufficiently impactful to advance knowledge in this field, thus the manuscript does not meet a criterion of Nature Communications.

I suggest two options. Considering high quality of their experiments, submitting the manuscript to other journals that are more focused on specific topics such as nanomaterials and nanotechnology would be recommended. Another option is to resubmit the manuscript after involving experimental proof of an impactful consequence of the embedded nanocluster formation, for example, drastic improvement in stability and efficiency in catalytic activities of Au films owing to the formation of embedded nanoclusters.

We are pleased that the reviewer finds that our manuscript is a beautiful work containing state-of-the-art measurements. We are also pleased that the reviewer finds our results interesting and novel. However, the reviewer has doubts about the level of impact. His/her doubts are mainly related to the fact that our present results can be understood largely from previous publications. We believe that we did not sufficiently point out the remarkable differences with previous results.

In fact we believe that at least part of the reviewer's concern is related to our use of the well-known particle-in-a-box model for the analysis of our new results. The use of the particle-in-a-box model to interpret standing wave patterns in STM experiments is obviously not new and well documented for various systems. Our main finding related to the use of the particle-in-a-box model is not that the observed standing wave patterns are consistent with the model, yet that the electrons in the SFT are electronically decoupled from its metal surroundings, which is an original use of the model. Our conclusion on the decoupling stems from the fact that the electrons are described by an effective mass and onset energy that is drastically different from that of the bare Au(111) surfaces state. Our observation of strongly decoupled (truncated) stacking-fault tetrahedra that are embedded in a metal film is unexpected and unique and was not reported in our previous work on Ag SFTs in Ag(111) films. To date, the local electronic properties of individual SFTs have remained unexplored. We are therefore convinced that our present finding is sufficiently impactful and will spark further research on this matter.

In the revised manuscript, we have modified the paragraphs related to the comparison to previous literature to better emphasize the novelty of our findings. In addition we now discuss in more detail the impact of our findings towards possible applications at the end of the manuscript. We hope that the reviewer will find our revised manuscript suitable for publication in *Nature Communications*.

The authors studied the electronic properties of stacking fault nanocrystals on Au(111) thin films. Au(111) thin films consisting of atomically flat islands were prepared by molecular beam epitaxy followed by repeated Ar ion bombardment and annealing. From detailed TEM, STM and STS characterizations, the nanocrystals were found to show a set of discrete electronic resonances and display pronounced voltage-dependent wave patterns in the maps of the density of states. The wave patterns exist up to the energies above the bottom of the bulk conduction band of the Au(111) film, indicating that the electronic properties of the Au nanocrystals are decoupled from its surrounding. This behavior was also found to correlate with the size of the Au nanocrystals. The experimental results were further analyzed by using a two-dimensional particle-in-a-box model, which matches well with their experiments.

In general, this paper is well organized and convincing. The topic and the results are novel in this filed and will be helpful for understanding the electronic properties of metal nanostructures. Therefore, I would recommend its publication on Nature Communications after some minor revisions.

We are grateful to the reviewer for his/her appreciation of our work and for his/her remarks that help us to further improve the manuscript. Below we reply in detail to his/her remaining remarks. We hope that the reviewer will find our revised manuscript suitable for publication in *Nature Communications*.

Below are some detailed comments.

1. The investigation methods and results of this work look quite similar to those for Ag SFTs on Ag(111) (a previous work from the authors, 2012 Phys. Rev. Lett. 108, 076806). I suggest the authors to emphasize the novelty and significance of this work, comparing with their previous work, in the introduction part.

Following the remark of this reviewer and that of reviewer #2, we compare in more detail the present work with the indicated previous work in the introduction part of the revised manuscript.

2. What is the difference or relationship between the two-dimensional image potential states reported here and the multipolar plasmonic states reported in larger crystalline nanoprisms (2013 Nat. Mater. 12, 426)?

Surface plasmons can be described as a collective oscillation of the density of (free) electrons at the surface and they can appear as patterns in the LDOS of nanoparticles as demonstrated in the article indicated by the reviewer, and also, *e.g.*, in *Sci. Rep.* **5**, 16635 (2015). In contrast, the patterns formed by both image-potential states (ISs) and surface states (SSs) are related to the quantum mechanical wave-like nature of electrons and the patterns arise due to interference between scattered electron waves.

SS electrons and plasmons are located *in* the surface and typically exist at energies well below the vacuum level, while ISs exist at high energies close to the vacuum level *above* the surface (in the presence of an electric field they are shifted typically above the vacuum level). In particular, IS electrons are confined along the surface normal by the crystal surface potential at one side and by the Coulomb-like image potential at the vacuum side. This results in the formation of hydrogen-like states, in which electrons act as a two-dimensional free-electron-like gas that can move freely parallel to the surface. The interaction of IS electrons with underlying bulk and surface state electrons causes the ISs to broaden into resonances. This

interaction involves scattering of the electrons (with other electrons or phonons), after which the electrons can decay to bulk states or SSs. The coupling of the electrons to the system can also involve (surface) plasmons, *i.e.*, plasmon excitations can partially determine the decay rate of the IS electrons.

In the revised manuscript, we added two sentences to explain the origin of image-potential states, which was lacking in the first version of the manuscript.

3. In Figure 5, why are the error bars of the resonances B and C not shown?

The error bars of the resonances B and C are the same as that for resonance A. This is now mentioned in the revised version of the manuscript.

4. In Figure 1a, besides the two indicated defects that appear at "crossroads", there are other defects that are not located at the "crossroads". For the HRTEM image, there are no evidences to show that the defect observed on the TEM image is the defect at "crossroads".

The reviewer is correct that other defects also exist in Fig. 1a. These defects are commonly observed in STM topographies of our Au(111) films and are interpreted as (sub)surface atomic-size defects (*e.g.*, an impurity atom or a missing Au atom) in the top atomic layers of the Au(111) film. These defects act as effective scattering centers for both surface and bulk electrons, which we have previously reported in Ref. [Scho2009a]. The defects at the "crossroads" are clearly larger and consisting of several tens of atoms, which is comparable to previous STM and TEM observations of SFTs in Ag(111) films. It is therefore reasonable to link the Au SFTs observed in the TEM images to these larger defect structures at the "crossroads".

In the revised manuscript we point to the existence of the atomic-size defects in Fig. 1a.

5. In Figure 5, the theoretical plots are obviously deviated from the experimental results, especially for the resonances B and C. The authors had better give some explanation for it.

The theoretically expected size dependence that is plotted in Fig. 5 is that for the case of a hexagonal box, which is an assumed and idealized shape for that of the exposed facet of the SFT. Deviations of the experimental data in Fig. 5 from the theoretical curve may be attributed at least partially to deviations from the assumed ideal hexagonal (truncated triangular) shape.

For the larger SFTs in Fig. 5 the agreement is very reasonable, while the main deviations occur for the smaller SFTs in Fig. 5. For the larger SFTs such as that in Fig. 1a, Fig. 2a and Fig. 3a, the truncated triangular shape can be observed quite clearly in the STM topographies. For the smaller SFTs such as that in Fig. 4a, the precise shape can be observed less clearly due to the more dominating tip convolution effects. Their shape in fact may be more close to that of a triangle rather than a hexagon.

In addition, electron scattering at the stacking fault planes of the SFT below the surface may further influence the ideal particle-in-a-box type confinement of electrons.

Following the remark of the reviewer, we now discuss explicitly the deviations observed in Fig. 5.

6. *The dI/dV spectra of the Au NCs with the triangular shape are clearly seen to be different from those of the Au NCs with the hexagonal shape. The authors should provide discussion on this difference.*

The spectrum of the triangularly shaped defect in Fig. SM-4 indeed strongly differs from that of the other SFTs discussed in the manuscript. The spectrum shows a pronounced resonance that is shifted towards the Fermi level when compared to the resonance observed on the surrounding Au(111) surface. The resonance on the Au(111) surface reflects the onset of the Au(111) surface state that occurs around -460 meV. In addition, wave patterns can already be observed in LDOS maps at voltages close to the Fermi level (Fig. SM-4f and SM-4g), while the SFTs discussed in the manuscript only show wave patterns above 1490 mV. The spectrum of the triangular defect and the LDOS maps in Fig. SM-4 can be interpreted in terms of the 2D particle-in-box model with $E_0 = -460$ meV and $m^* = 0.23 m_e$, similar to the results reported in Ref. [Scho2009b] for Au islands.

As mentioned above, the quantum confinement behaviour of the triangular defect is very different from that of the truncated triangular defects (SFTs) that exist at the crossroads of three herringbones exclusively. In particular, modelling of the SFT quantum behaviour using the 2D particle-in-a-box model reveals a very different onset energy and electron effective mass, *i.e.*, $E_0 = 1490$ meV and $m^* = 0.33 m_e$. Moreover, the triangular defect does reveal the onset of the Au(111) bulk conduction band, in contrast to the (truncated) SFTs. As pointed out by reviewer #1, the triangular feature may be interpreted as a so-called Frank loop, which is a 2D-type SF.

In the revised manuscript, we have rephrased the related paragraph to better emphasize our reasoning.

7. *I do not know the potential applications of these electrically decoupled stacking fault nanocrystals embedded in Au(111) films. Could the authors provide some discussion on this aspect.*

Because of their rich electronic properties that are related to the emergence of quantum size effects, SFTs may offer yet unexploited potential for new electronic, optical and chemically sensitive thin-film based devices. A possible route toward potential applications could be the use of SFTs as quantum dots in the metal films. SFTs that are embedded in metal films likely can be expected to have an enhanced stability compared to quantum dots grown on thin insulating films. *E.g.*, deposited molecular adsorbates on thin insulating films such as NaCl typically remain stable only when deposited on the cold substrate. Previously we found that also deposited Au clusters on NaCl films show some degree of mobility on the room temperature substrate, and they are often difficult to image with the STM tip [*Nano Lett.* **16**, 3063–3070 (2016)]. Moreover, the deposited clusters show some degree of deformation upon landing on the surface. The Au cluster that comes most close to our investigated Au SFTs is the known pyramidal Au₂₀ cluster. Again the stability of the deposited Au₂₀ cluster is rather limited [*Nanoscale* **4**, 4947 (2012)], while SFTs are very robust at room temperature. In addition, formation of SFTs in magnetic materials may open up possibilities for magnetic applications. Obviously, their controlled preparation will be a crucial issue for further developments in these directions. A potential route to overcome this issue could be by creating regular patterns of defects in the substrate, at which SFTs may preferentially start to form during the metal film growth on the support.

Besides the speculative impact discussed above, at the present stage it is already evident from our findings that the investigation of the electronic properties of SFTs opens up a new playground for fundamental investigations of quantum mechanical finite size effects in surfaces that deserves further experimental and theoretical exploration.

In the revised manuscript, we elaborate more on the interest of SFTs for further investigations and hereby added a reference to *Nanoscale* **4**, 4947 (2012) and to our previous work *Nano Lett.* **16**, 3063–3070 (2016).

Reviewers' comments:

Reviewer #1 (Remarks to the Author):

This manuscript is unacceptable for the specialists of irradiation damage unless otherwise the authors provide clear evidence that the SFT is not the artifact of FIB damage. The authors claim that FIB introduces no irradiation damage into samples. Their idea is against a general consensus of the research community of irradiation damage. FIB certainly introduces irradiation damage; either flash electro-polishing or gentle milling with $\ll 1$ keV Ar ions are mandatory required for eliminating the FIB damage. Since Nature Communications is a highly influential journal, if this manuscript was published in this journal, many young scientists will follow the authors' logic by using this article as a reference. That will bring an unnecessary confusion to the irradiation damage research community. Indeed, this subject is delicate for us.

The detection limit of EDS is no better than 1 at.%. Given that a single Ga ion introduces a single vacancy (though this assumption is too conservative without considering the effect of cascade damage), the resultant local vacancy concentration in the sample is 1 at.%, which is one or two orders of magnitude greater than the thermal equilibrium vacancy concentration just below the melting point.

The FIB damage microstructure strongly depends on materials, more specifically, the mobility of vacancies. Vacancy clusters do not form in aluminum, in which vacancies are highly mobile at room temperature; all vacancies escape to the free surface. Vacancy clusters do not form in many intermetallic compounds, in which vacancies are immobile at room temperature. Transmission electron microscopy observation is incapable of detecting single vacancies. Vacancy clusters are favored to form in gold and steels, in which the mobility of vacancies is moderate under irradiation at room temperature via irradiation-enhanced diffusion.

Flash electro-polishing is not limited to steels but applicable to any metals. Gold is easier than steels as the thinning rate of electro-polishing is much lower. It takes just a minute for electro-polishing a 3 mm TEM disc of steels by twin-jet polisher, but takes more than thirty minutes for a gold disc. The basic procedure of flash electro-polishing follows classic 'window polishing' rather than twin-jet polishing. The recipe of window polishing for gold is as follows.

Solution: Distilled water 200 cc
Potassium cyanide 13.6 g
Potassium ferrocyanide 3 g
Potassium sodium tartrate 3 g
Phosphoric acid 4 g
Ammonium hydroxide (25%) 0.8 g
Temperature: ~ 276 K (cool down with ice water)
Voltage: ~ 5 V (dependent on the distance between sample and electrodes)

The experiment of fully annealed gold samples is no difficult; small bulk gold samples are available for purchase.

Small vacancy clusters are often invisible in high-resolution images. In thick foils (e.g. 30 nm) the signal from SFTs (2 nm) is hindered by the signal from the matrix; SFTs are not detectable even by using through-focus imaging. In such a case, further validation using diffraction contrast imaging is mandatory for deriving a conclusion that SFTs are absent. When the sample thickness is sufficiently thin for detecting SFTs in HR-TEM images (e.g. 10 nm foils), FIB artifact would be unavoidable.

The results of numerical calculations are highly dependent on the initial assumptions such as the damage threshold energy, which is ambiguous for many materials. For instance, the critical accelerating voltage of electron irradiation causing damage production in aluminum was believed to be 160 kV fifty years ago from now; later, the critical voltage was updated to be 120 kV based on empirical knowledge obtained by experiments. Likewise, the critical accelerating voltage of electron irradiation causing damage production in carbon nanotubes was believed to be 80 kV in the early 2000s; several years ago the critical voltage was updated to be 60 kV based on empirical knowledge obtained by experiments. This is the main reason that the lowest accelerating voltage of FEI Titan series was updated from 80 kV (the first generation) to 60 kV (the second generation). Based on these experiences, we are aware of the vulnerability of numerical calculations; experimental assessment is the first priority for this delicate discussion.

Reviewer #3 (Remarks to the Author):

I read through the response made by the authors and checked the revised sentences in the manuscript. I think the response made by the authors is very reasonable. The metallic quantum dots themselves look very interesting to me, although it might be hard to come up with some applications at this stage. I would recommend the publication of this manuscript on Nature Communications.

Reply to the comments from the reviewers

Reviewer #1:

- (1) This manuscript is unacceptable for the specialists of irradiation damage unless otherwise the authors provide clear evidence that the SFT is not the artifact of FIB damage. The authors claim that FIB introduces no irradiation damage into samples. Their idea is against a general consensus of the research community of irradiation damage. FIB certainly introduces irradiation damage; either flash electro-polishing or gentle milling with $<<1$ keV Ar ions are mandatory required for eliminating the FIB damage. Since Nature Communications is a highly influential journal, if this manuscript was published in this journal, many young scientists will follow the authors' logic by using this article as a reference. That will bring an unnecessary confusion to the irradiation damage research community. Indeed, this subject is delicate for us.

*Reply: we never said that FIB cannot introduce artifacts, what we said is that we have taken all precautions possible to avoid the occurrence of pronounced artifacts such as the unwanted creation of SFTs. We are convinced that we are not sending the wrong message to any reader. On the contrary, we clearly indicate that people need to be cautious about this. In concreto, under Methods we write "In order to **minimize** any damage on the sample during thinning ..." which directly implies that damage can be expected and that we need to be careful about it.*

- (2) The detection limit of EDS is no better than 1 at.%. Given that a single Ga ion introduces a single vacancy (though this assumption is too conservative without considering the effect of cascade damage), the resultant local vacancy concentration in the sample is 1 at.%, which is one or two orders of magnitude greater than the thermal equilibrium vacancy concentration just below the melting point.

Reply: This is generally speaking correct, but again we do not claim that there is no Ga in the sample, we wrote "Energy Dispersive X-Ray (EDX) analysis was performed in TEM and did not reveal any trace of Ga⁺ ions ...". We have now extended this by saying "However, with a detection limit of around 1 at.% this does not exclude the existence of some Ga in the film and hence the production of extra vacancies, ..." which is then followed by the response to point (5).

- (3) The FIB damage microstructure strongly depends on materials, more specifically, the mobility of vacancies. Vacancy clusters do not form in aluminum, in which vacancies are highly mobile at room temperature; all vacancies escape to the free surface. Vacancy clusters do not form in many intermetallic compounds, in which vacancies are immobile at room temperature. Transmission electron microscopy observation is incapable of detecting single vacancies. Vacancy clusters are favored to form in gold and steels, in which the mobility of vacancies is moderate under irradiation at room temperature via irradiation-enhanced diffusion.

Reply: Again correct, but irradiation damage does occur in most metals, so any technical

experience during FIB thinning used to avoid this damage will be beneficial when used for other systems.

- (4) Flash electro-polishing is not limited to steels but applicable to any metals. Gold is easier than steels as the thinning rate of electro-polishing is much lower. It takes just a minute for electro-polishing a 3 mm TEM disc of steels by twin-jet polisher, but takes more than thirty minutes for a gold disc. The basic procedure of flash electro-polishing follows classic 'window polishing' rather than twin-jet polishing. The recipe of window polishing for gold is as follows.

Solution: Distilled water 200 cc

Potassium cyanide 13.6 g

Potassium ferrocyanide 3 g

Potassium sodium tartrate 3 g

Phosphoric acid 4 g

Ammonium hydroxide (25%) 0.8 g

Temperature: ~276 K (cool down with ice water)

Voltage: ~5 V (dependent on the distance between sample and electrodes)

Reply: Of course this method can be applied to any metal. However, the conditions for window polishing are not necessarily the same as for the flash polishing. Moreover, one can never completely convince the reader, since one never knows exactly how much material has been removed while one also does not know how deep the FIB artifacts had been created, if any. If the SFTs are indeed FIB artifacts, as of some length of time for the flash polishing they would indeed disappear and a clean sample would be observed (assuming enough material is left for study). However, if the SFTs are indeed genuine to the sample, i.e. not produced by the FIB, one will always see remaining SFTs and a reviewer will always be able to claim that the flash-polishing was not performed long or severe enough. In other words, this experiment can take a very long time to perform (ideally it asks for several FIB samples and several runs to cover a large space of thinning parameters) but can never for 100% disprove that the SFT are FIB artifacts. (see also further below) The same holds for gentle Ar milling also proposed above by the reviewer under (1). Again, since from STM we are sure the SFT exist, we would always be in the situation that SFT remain visible and anyone can claim the flash-polishing was insufficient.

- (5) The experiment of fully annealed gold samples is no difficult; small bulk gold samples are available for purchase.

Reply: following the suggestion of the reviewer we now annealed a 99.99% gold bulk sample for 24h at 700°C, which should remove all one- and two-dimensional defects (see e.g. [1]). From this piece a FIB sample (further referred to as "pristine Au sample") was produced using the same conditions as for the Au film of the manuscript. Below three typical HRTEM images from different regions are shown. In the first no irradiation damage can be observed, in the second the signature of a dislocation loop can be recognized and in the third a single stacking fault with strain field is seen. However, over the entire size of the FIB sample (approximately 3 by 3 micron²)

no signature of a SFT was seen, regardless the defocus value used. In other words, the FIB conditions used on a pristine Au sample did not produce any visible SFTs, since these would have been visible under similar conditions as for the visible single stacking faults. This implies that the FIB thinning indeed produces defects, as already indicated before, but not the investigated SFT. In order to fully inform the reader, these images are now also provided in the supplementary material.

[1] J. H. CHO et al., *Metallurgical and materials transactions A*, 36 (2005) 3415.

Beside this approach, the original film was also peeled off from the substrate and investigated as such, i.e. without further FIB thinning. Of course, a film with an average thickness of 140 nm will not produce proper HRTEM images, but still contrasts as shown below and resembling the shape of a SFT could be observed. An attempt was made to thin this peeled film by flash-electropolishing, but without success (the material immediately disappeared, again showing the difficulty of this approach (see also point (4)). Thinner films on mica with sufficient conductivity for the electrical tests could not be produced, so any other direct measure via TEM does not seem feasible.

- (6) Small vacancy clusters are often invisible in high-resolution images. In thick foils (e.g. 30 nm) the signal from SFTs (2 nm) is hindered by the signal from the matrix; SFTs are not detectable even by using through-focus imaging. In such a case, further validation using diffraction contrast imaging is mandatory for deriving a conclusion that SFTs are absent. When the sample thickness is sufficiently thin for detecting SFTs in HR-TEM images (e.g. 10 nm foils), FIB artifact would be unavoidable.

Reply: Again, the approach of the reviewer is a bit strange. Our assumption is that we are seeing SFTs which are expected in the material based on the STM observations. So we do not need to prove that they are there. As shown above, we did not see any SFT in the pristine Au sample after FIB thinning, so if FIB still did introduce very small SFTs in the film, we would not have seen those either, implying these would not have cluttered our interpretation. SFTs as those observed in the FIBbed film do not appear in the pristine Au sample.

One note on the visibility under HRTEM conditions: the image (c) shown under point (5) proves that SF contrast can be seen, even for very small SFs, the point is indeed to select the proper objective aperture and defocus value. So the same would hold for SFTs, they can be made visible even when being very small (as shown in Fig. 1 in the manuscript).

- (7) The results of numerical calculations are highly dependent on the initial assumptions such as the damage threshold energy, which is ambiguous for many materials. For instance, the critical accelerating voltage of electron irradiation causing damage production in aluminum was believed to be 160 kV fifty years ago from now; later, the critical voltage was updated to be 120 kV based on empirical knowledge obtained by experiments. Likewise, the critical accelerating voltage of electron irradiation causing damage production in carbon nanotubes was believed to be 80 kV in the early 2000s; several years ago the critical voltage was updated to be 60 kV based on empirical knowledge obtained by experiments. This is the main reason that the lowest accelerating voltage of FEI Titan series was updated from 80 kV (the first generation) to 60 kV (the second generation). Based on these experiences, we are aware of the vulnerability of numerical calculations; experimental assessment is the first priority for this delicate discussion.

Reply: We agree with the reviewer, and we feel we did everything possible to provide experimental evidence that the SFT are not produced by FIB and that we are simply observing with TEM the defects in the bulk that are also seen by the STM at the surface of non-FIBbed samples.

Reviewer #3:

I read through the response made by the authors and checked the revised sentences in the manuscript. I think the response made by the authors is very reasonable. The metallic quantum dots themselves look very interesting to me, although it might be hard to come up with some applications at this stage. I would recommend the publication of this manuscript on Nature Communications.

Reply: We sincerely appreciate the recommendation by the reviewer.